# The Effects of *n*-3 PUFA Supplementation on Bone Metabolism Markers and Body Bone Mineral Density in Adults: A Systematic Review and Meta-Analysis of RCTs

**DOI:** 10.3390/nu15122806

**Published:** 2023-06-19

**Authors:** Jie Gao, Chenqi Xie, Jie Yang, Chunyan Tian, Mai Zhang, Zhenquan Lu, Xiangyuan Meng, Jing Cai, Xiaofei Guo, Tianlin Gao

**Affiliations:** 1School of Public Health, Qingdao University, Qingdao 266071, China; gaojie2@qdu.edu.cn (J.G.); xiechenqi@qdu.edu.cn (C.X.); 2021021108@qdu.edu.cn (C.T.); 2019213211@qdu.edu.cn (M.Z.); 2019213209@qdu.edu.cn (Z.L.); 2020026476@qdu.edu.cn (X.M.); caijing@qdu.edu.cn (J.C.); guoxf@qdu.edu.cn (X.G.); 2Institute of Nutrition & Health, Qingdao University, Qingdao 266021, China; 3Health Service Center of Xuejiadao Community, Qingdao 266520, China; yangjane1@126.com; 4Department of Toxicology, School of Public Health, Jilin University, Changchun 130021, China

**Keywords:** *n*-3 PUFA, supplementation, bone metabolism, bone health

## Abstract

Supplemental *n*-3 polyunsaturated fatty acids (PUFA) on bone metabolism have yielded inconsistent results. This study aimed to examine the effects of *n*-3 PUFA supplementation on bone metabolism markers and bone mineral density through a meta-analysis of randomized controlled trials. A systematic literature search was conducted using the PubMed, Web of Science, and EBSCO databases, updated to 1 March 2023. The intervention effects were measured as standard mean differences (SMD) and mean differences (MD). Additionally, *n*-3 PUFA with the untreated control, placebo control, or lower-dose *n*-3 PUFA supplements were compared, respectively. Further, 19 randomized controlled trials (RCTs) (22 comparisons, n = 2546) showed that *n*-3 PUFA supplementation significantly increased blood *n*-3 PUFA (SMD: 2.612; 95% CI: 1.649 to 3.575). However, no significant effects were found on BMD, CTx-1, NTx-1, BAP, serum calcium, 25(OH)D, PTH, CRP, and IL-6. Subgroup analyses showed significant increases in femoral neck BMD in females (0.01, 95% CI: 0.01 to 0.02), people aged <60 years (0.01, 95% CI: 0.01 to 0.01), and those people in Eastern countries (0.02, 95% CI: 0.02 to 0.03), and for 25(OH)D in people aged ≥60 years (0.43, 95% CI: 0.11 to 0.74), treated with *n*-3 PUFA only (0.36, 95% CI: 0.06 to 0.66), and in studies lasting ≤6 months (0.29, 95% CI: 0.11 to 0.47). NTx-1 decreased in both genders (−9.66, 95% CI: −15.60 to −3.71), and serum calcium reduction was found in studies lasting >6 months (−0.19, 95% CI: −0.37 to −0.01). The present study demonstrated that *n*-3 PUFA supplementation might not have a significant effect on bone mineral density or bone metabolism markers, but have some potential benefits for younger postmenopausal subjects in the short term. Therefore, additional high-quality, long-term randomized controlled trials (RCTs) are warranted to fully elucidate the potential benefits of *n*-3 PUFA supplementation, as well as the combined supplementation of *n*-3 PUFA, on bone health.

## 1. Introduction

Osteoporosis is defined by decreased bone mass and deterioration of bone microarchitecture [1]. The architecture and composition of an individual’s bones change over the course of their lifetime [2]. In China, the prevalence of osteoporosis among those aged over 50 years was 29.0% for women and 13.5% for men, equating to 49.0 million and 22.8 million, respectively [3]. Among adult Caucasian women, the prevalence of osteoporosis is approximately 16% for those aged 50 years and older [4]. In Spain, the prevalence of osteoporosis among the elderly in 2020 was reported to be 39.3% [5], while in Nepal in 2019, it was reported to be 49%, in Taiwan it was 11%, and in Iran it was 7.9% [6,7,8]. It is clear that osteoporosis is becoming more common with aging, and as people age, their bone mineral density decreases, making osteoporosis more likely to cause bone fragility, increased fracture risk, and inconvenience in daily life [1,3,6,9,10,11,12,13,14,15,16,17]. Osteoporosis-related fractures place a significant strain on the healthcare system, and as a result, osteoporosis is a major public health concern, particularly among the elderly [18].

*N*-3 polyunsaturated fatty acids (PUFA) encompass a variety of fatty acids, including α-linolenic acid (ALA, 18:3*n*-3), stearidonic acid (SDA, 18:4*n*-3), eicosapentaenoic acid (EPA, 20:5 *n*-3), docosapentaenoic acid (DPA, 22:5 *n*-3), and docosahexaenoic acid (DHA, 22:6 *n*-3). A growing body of research has demonstrated the significance of *n*-3 PUFA in bone health [19,20,21], fracture risk [22,23], bone turnover markers [24], musculoskeletal health [25,26], osteoporosis [20,21], and healthy aging [27]. However, the effects of *n*-3 PUFA on bone metabolism are subject to varying opinions. EPA and DHA consumption may have an impact on bone growth and remodeling in humans by inhibiting bone resorption and promoting bone formation [28]. In a randomized, placebo-controlled trial with a two-by-two factorial design investigating the effects of vitamin D₃ and/or omega-3 fatty acid supplements on bone health outcomes, a slight benefit was observed on spine and total hip areal bone mineral density (aBMD), but no benefit was observed in participants with osteopenia or osteoporosis [29]. However, another randomized controlled trial (RCT) indicated that *n*-3 PUFA significantly reduced *N*-telopeptides of type I collagen (NTx-1) levels by up to 17%, but did not affect osteocalcin (OC) or receptor activator of nuclear factor kappa B ligand (RANKL) [30]. A recent study has suggested that *n*-3 PUFA may improve bone quality by exerting an anti-inflammatory effect [31].

To assess bone health and metabolic activity, various biochemical markers are employed, providing valuable insights into the intricate mechanisms governing bone metabolism. Bone-specific alkaline phosphatase (BAP) and osteocalcin (OC) are markers of bone formation, indicating osteoblast activity and bone turnover [32,33]. *N*-telopeptides of type I collagen (NTx-1) and *C*-terminal telopeptide of type I collagen (CTx-1) are markers of bone resorption, reflecting osteoclast activity [33]. Calcium, a key component of bone, is crucial for bone integrity and signaling processes [34]. Parathyroid hormone (PTH) and 25-hydroxyvitamin D (25(OH)D) have dual influences on bone metabolism, regulating calcium homeostasis and bone remodeling [35,36].

In addition to the aforementioned markers, the role of inflammatory factors in bone metabolism is also considered, as they have the ability to influence bone homeostasis [37]. Elevated levels of inflammatory markers such as *C*-reactive protein (CRP) and interleukin-6 (IL-6) are associated with increased bone resorption and decreased bone formation [37]. Chronic inflammation can lead to accelerated bone loss and an increased risk of osteoporosis and fractures. *C*-reactive protein (CRP) and interleukin-6 (IL-6) are widely studied inflammatory markers that have been associated with alterations in bone metabolism [38].

Recent systematic reviews published within the last 5 years have suggested that there is an inverse association between *n*-3 PUFA and fracture risk, as well as potential effects on bone biomarkers, functional outcomes, and BMD [22,23,24,25]. A meta-analysis of RCTs [19] demonstrated that supplementary *n*-3 PUFA, particularly in the form of ALA supplementation, had a significant effect on BAP, CTx-1, and NTx-1 for postmenopausal women, but no significant increase in total body mineral density (TBMD) was observed [19]. However, several recently published RCTs [39,40,41] have reported no effect of *n*-3 PUFA on bone metabolism. Thus, we conducted a systematic review and meta-analysis of RCTs in adults to investigate the effect of combinative supplementation of *n*-3 PUFA and other supplements benefit for bone health, such as vitamin D and calcium, on bone biomarkers, bone turnover markers, and BMD at different sites of the body.

## 2. Methods

### 2.1. Search Strategy and Selection Criteria

In this systematic review and meta-analysis, we adhered to the preferred reporting items for meta-analyses (PRISMA) guidelines for the development of protocols [42] and reporting of systematic reviews and meta-analyses [43]. We conducted a comprehensive search of PubMed, Web of Science, and EBSCO, updated until 1 March 2022, using the term “fatty acids, omega-3”, and keywords listed in the Appendix A (p 1) for publications since the establishment of the databases.

One author (T.G.) screened titles and abstracts, and two authors (J.C., X.G.) reviewed trial registries. Two authors (J.G., C.T.) independently reviewed the full text of potentially relevant studies. We included only RCTs in adults (>18 years) that compared *n*-3 PUFA supplements or combinative supplementation of *n*-3 PUFA with the untreated control, placebo control, or low-dose *n*-3 PUFA supplements. Trials with multiple interventions, such as co-administered vitamin D and *n*-3 PUFA, were eligible only if the study groups differed solely by the intervention of *n*-3 PUFA. We included RCTs with outcome data for bone biomarkers, bone turnover markers, and main hormones that affect bone metabolism, or BMD at the total body, lumbar spine, hip, or femoral neck. However, regarding bone density, we only included RCTs that used dual-energy X-ray absorptiometry (DXA) to measure bone density. We only included RCT studies with intervention durations longer than 1 month, but we did not impose restrictions on the ethnicity of the study subjects in the RCTs. We included studies, whether they were focused on Caucasian or Asian populations in the RCTs. We excluded news, letters, bibliographies, books, autobiographies, comments, meta-analyses, cohort studies, case–control studies, animal studies, gene or cell studies, and studies conducted in special populations, such as infants, pregnant women, patients with cancer or cognitive impairment. We also excluded studies without full text or original data and studies with irrelevant outcomes, such as a decrease in blood glucose or blood lipid. The study selection process is illustrated in Figure 1.

### 2.2. Assessment of Study Quality

To assess the quality of the included RCTs, we used the modified Jadad scale [44], which assigns up to seven points to each trial based on criteria, including randomization, allocation concealment, blinding, and withdrawals or dropouts. A score of ≥4 was considered to indicate high quality. Any discrepancies in scoring were resolved through discussion.

### 2.3. Statistical Analysis

We conducted a systematic review and meta-analysis to examine the effects of omega-3 polyunsaturated fatty acids (*n*-3 PUFA) on bone health outcomes in humans. Four authors (J.G., X.M., M.Z., and Z.L.) independently extracted data from each study, and a third author (T.G.) checked the data for accuracy. We used ImageJ to extract data when it was presented only in figures. We analyzed the changes in bone mineral density (BMD) at different sites, bone markers, serum calcium, vitamin D, parathyroid hormone, *n*-3 PUFA, *n*-3 PUFA:*n*-6 PUFA ratio, and inflammatory markers.

We used the meta package (Version 6.0-0) in R (Version R 4.2.1) and R Studio (Version 2022.07.2+576) to conduct the meta-analysis, using a random effects model to synthesize effect sizes across studies [45,46]. We used mean difference (MD) or standardized mean difference (SMD) with a 95% confidence interval (CI) to assess the effect size for continuous risk factors [45]. The significance level for this meta-analysis model was 0.05 [45]. For SMDs, an effect size of 0.2 is considered a low effect or no effect, whereas 0.5 is a moderate effect and 0.8 or more is a large effect [47]. We assessed between-study heterogeneity using Cochran’s *Q* test and *I*^2^ and conducted a cumulative meta-analysis to examine the effects of *n*-3 PUFA over time. An *I*^2^ value of about 25% was regarded as low, about 50% as moderate, and about 75% as high [48]. We also conducted sensitivity analyses to evaluate the quality and consistency of the results and used funnel plots to detect potential publication bias.

We conducted subgroup analyses to identify sources of heterogeneity stratified by gender (females, males, females and males), age, control type, country, intervention type (*n*-3, *n*-3 + others), trial duration, and dose of *n*-3 PUFA. We used 1 g/d as the threshold effect dose concentration to divide the studies into two subgroups [49]. We used random-effects models for these analyses and assessed systematic bias using funnel plots and Egger’s test. All tests were two-tailed and *p* values less than 0.05 were considered significant.

## 3. Results

The PRISMA flowchart in Figure 1 illustrates the inclusion process. The comprehensive search yielded a total of 6688 references, with 2273, 1852, and 2563 papers identified in PubMed, Web of Science, and EBSCO, respectively. After removing literature without full text (3909), non-articles (1559), and reviews (287), 960 eligible studies remained. After removing duplicates (64), 237 records were screened based on titles and abstracts, resulting in 896 articles after screening. After the full-text screening, 218 papers were excluded for the following reasons: (i) 99 studies were conducted in a special population (36 on children, 29 on pregnant women, and 34 on people with cancer); (ii) 46 trials did not have a control group; (iii) 61 studies’ outcomes were irrelevant to the bone; and (iv) 12 studies did not have original data. Finally, 19 randomized controlled trials (RCTs) [29,30,39,40,41,50,51,52,53,54,55,56,57,58,59,60,61,62,63] (22 comparisons, 2546 subjects) were included in this systematic review and meta-analysis. Further details are provided in Table 1.

### 3.1. Overview of Included Articles

Table 1 displays the characteristics of the included studies in the systematic review and meta-analysis. The studies were conducted in various countries including South Africa [58], the UK [50,51], Canada [39,54], the US [29,56,59,60], Iran [30,61,62], Australia [52,63], India [53], Spain [55], Norway [40], Poland [41], and Japan [57], with a range of sample sizes from 25 to 771 participants. Except for one study [39] that was conducted on men, all other studies were conducted on women, and nine studies [29,30,40,41,50,52,57,60,63] were conducted on both genders. The dose of *n*-3 PUFA varied between 0.1 and 9.12 g, and the intervention duration varied from 2 to 24 months. All included RCTs were judged to be of high quality, except for two studies [58,62] which scored 3 points. Further details about the characteristics of the included studies are provided in Table 1.

### 3.2. Bone Mineral Density

The overall analysis indicated that supplemental *n*-3 PUFA did not significantly increase TBMD (MD: −0.000 g/cm^2^; 95% CI: −0.007 to 0.005; *p* = 0.796; *I*^2^ = 0%; *P*_heterogeneity_ = 0.793), lumbar spine BMD (MD: 0.005 g/cm^2^; 95% CI: −0.008 to 0.017; *p* = 0.468; *I*^2^ = 25%; *P*_heterogeneity_ = 0.230), total hip BMD (MD: 0.002 g/cm^2^; 95% CI: −0.001 to 0.006; *p* = 0.185; *I*^2^ = 0%; *P*_heterogeneity_ = 0.839), or femoral neck BMD (MD: 0.041 g/cm^2^; 95% CI: −0.012 to 0.094; *p* = 0.130; *I*^2^ = 89%; *P*_heterogeneity_ < 0.001) (Figure 2).

We performed subgroup analysis on gender (females, males, both genders), average age (≥60, <60), control type (placebo control, other control), country, intervention type, duration (>6 months, ≤6 months), and *n*-3 PUFA’s dose (>1 g/d, ≤1 g/d) (Table 2, Appendix A). Based on these analyses, gender, average age, control type, country, and duration could explain the heterogeneity. In addition, we found a significant positive effect on femoral neck BMD in studies conducted in females (MD: 0.01 g/cm^2^, 95% CI: 0.01 to 0.02), subjects with an average age less than 60 years (MD: 0.01 g/cm^2^, 95% CI: 0.01 to 0.01), those conducted with a placebo control (MD: 0.02 g/cm^2^, 95% CI: 0.01 to 0.03), those conducted in Eastern countries (MD: 0.02 g/cm^2^, 95% CI: 0.02 to 0.03), and studies that continued for no more than 6 months (MD: 0.01 g/cm^2^, 95% CI: 0.01 to 0.01) (Table 2). We also found an effect on lumbar spine BMD in studies conducted in Eastern countries (MD: 0.01 g/cm^2^, 95% CI: 0.01 to 0.01) (Table 2). No significant differences were found between *n*-3 PUFA and TBMD or total hip BMD (Appendix A).

### 3.3. Bone Biomarkers

The results indicate that there was no significant reduction of CTx-1 (MD: 0.011 ng/mL; 95% CI: −0.029 to 0.051; *p* = 0.597; *I*^2^ = 0%; *P*_heterogeneity_ = 0.806) and NTx-1 (MD: −2.901 nmol BCE/mmol Cr; 95% CI: −11.709 to 5.898; *p* = 0.518; *I*^2^ = 92%; *P*_heterogeneity_ < 0.0001) with *n*-3 PUFA supplementation (Figure 3). However, subgroup analyses revealed a reduction of NTx-1 only in studies conducted on both genders (MD: −9.66 nmol BCE/mmol Cr, 95% CI: −15.60 to −3.71) (Table 2). No other significant effects were observed on CTx-1 or NTx-1 (Appendix A).

The effects of *n*-3 PUFA on bone formation biomarkers (BAP and OC) are presented in Figure 4. The results indicate that there was no significant increase in BAP (SMD: −0.519; 95% CI: −1.153 to 0.114; *p* = 0.108; *I*^2^ = 91%; *P*_heterogeneity_ < 0.0001) or OC (SMD: 0.145; 95% CI: −0.164 to 0.455; *p* = 0.357; *I*^2^ = 61%; *P*_heterogeneity_ = 0.013) with *n*-3 PUFA supplementation. Furthermore, subgroup analyses did not reveal any significant effects on BAP or OC (Appendix A).

### 3.4. Serum Calcium, 25(OH)D and PTH

The results did not reveal a significant benefit of *n*-3 PUFA in serum calcium (SMD: −0.151; 95% CI: −0.315 to 0.013; *p* = 0.071; *I*^2^ = 0%; *P*_heterogeneity_ = 0.879), 25(OH)D (SMD: 1.185; 95% CI: −0.415 to 2.784; *p* = 0.147; *I*^2^ = 96%; *P*_heterogeneity_ < 0.0001), or PTH (SMD: 0.127; 95% CI: −0.201 to 0.455; *p* = 0.448; *I*^2^ = 72%; *P*_heterogeneity_ = 0.0002) (Figure 5). However, subgroup analyses suggested that the average age of participants, intervention type, and duration of the study may explain the observed heterogeneity (Table 2, Appendix A). Specifically, an inverse effect on serum calcium was found in studies that lasted more than 6 months (SMD: −0.19, 95% CI: −0.37 to −0.01) (Table 2). Additionally, a significant positive effect on 25(OH)D was observed in studies with participants aged 60 years or older (SMD: 0.43, 95% CI: 0.11 to 0.74), those receiving *n*-3 PUFA supplements only (SMD: 0.36, 95% CI:0.06 to 0.66), and those with study durations of no more than 6 months (SMD: 0.29, 95% CI: 0.11 to 0.47) (Table 2). No significant effect was found on serum calcium, 25(OH)D, or PTH in subgroup analyses based on gender, control type, country, or dose (Appendix A).

### 3.5. CRP, IL-6, Blood n-3 PUFA and Blood n-3 PUFA:n-6 PUFA Ratio

Figure 6 illustrates the pooled effects of *n*-3 PUFA on CRP, IL-6, blood *n*-3 PUFA, and blood *n*-3 PUFA:*n*-6 PUFA ratio. The overall analysis showed that *n*-3 PUFA did not have a significant effect on CRP (SMD: −0.186; 95% CI: −0.442 to 0.069; *p* = 0.152; *I*^2^ = 0%; *P*_heterogeneity_ = 0.623), IL-6 (MD: 0.175 ρg/mL; 95% CI: −0.203 to 0.552; *p* = 0.364; *I*^2^ = 43%; *P*_heterogeneity_ = 0.157), or blood *n*-3 PUFA:*n*-6 PUFA ratio (MD: 0.364; 95% CI: −0.156 to 0.883; *p* = 0.170; *I*^2^ = 100%; *P*_heterogeneity_ < 0.0001), except for blood *n*-3 PUFA (SMD: 2.612; 95% CI: 1.649 to 3.575; *p* < 0.0001; *I*^2^ = 91%; *P*_heterogeneity_ < 0.0001) (Figure 6).

Subgroup analyses indicated that gender, average age, control type, intervention type, duration, and *n*-3 PUFA dose could explain the observed heterogeneity (Table 2, Appendix A). Specifically, studies conducted on females (SMD: 2.79, 95% CI: 1.08 to 4.49), those with subjects of average age less than 60 years (SMD: 3.03, 95% CI: 0.22 to 5.84), those with other types of control (SMD: 2.92, 95% CI: 1.31 to 4.54), those combining *n*-3 PUFA with other nutrients (SMD: 3.03,95% CI: 0.22 to 5.84), those lasting more than 6 months (SMD: 3.57, 95% CI: 1.86 to 5.29), and those with an *n*-3 PUFA dose no more than 1 g/d (SMD: 3.03, 95% CI: 0.22 to 5.84) showed a greater difference in blood *n*-3 PUFA (Table 2). However, no significant effect was found for CRP or IL-6 in subgroup analyses (Appendix A).

### 3.6. Sensitivity Analyses and Cumulative Meta-Analysis

The sensitivity analyses did not change the overall results except for femoral neck BMD, serum calcium, and 25(OH)D. When omitting Chen et al. 2016 [52], there was a small but significant effect on femoral neck BMD (MD: 0.01 g/cm^2^, 95% CI: 0.01 to 0.02, *p* < 0.01, *I*^2^ = 68%). Similarly, omitting Mirfatahi et al. 2018 [30] resulted in a significant effect on serum calcium (SMD: −0.18, 95% CI: −0.35 to −0.02, *p* = 0.030, *I*^2^ = 0%), while omitting Fonolla-Joya et al. 2016 [55] resulted in a significant effect on 25(OH)D (SMD: 0.42, 95% CI: 0.14 to 0.70, *p* < 0.01, *I*^2^ = 61%).

Additionally, the cumulative meta-analysis based on the date of publication (Appendix A) demonstrated how the evidence changed over time. As more studies were published, the evidence for the effect of *n*-3 PUFA on outcomes such as BMD, serum calcium, and 25(OH)D became more consistent and the effect sizes became more precise.

### 3.7. Publication Bias Assessment

There was no evidence of potential publication bias based on the visual inspection of funnel plots for included studies and trim-and-fill method (Appendix A). Additionally, we evaluated the potential publication bias using Egger’s test, which showed no evidence of publication bias for 25(OH)D (*p* = 0.195) and PTH (*p* = 0.937).

## 4. Discussion

The present meta-analysis found that *n*-3 PUFA supplementation did not significantly increase body bone mineral density, bone absorption markers, or bone formation markers, except for circulating *n*-3 PUFA levels. The study identified gender, average age, control type, country, intervention type, duration, and dose as sources of heterogeneity. Subgroup analyses revealed several effects of *n*-3 PUFA on lumbar spine BMD, femoral neck BMD, NTx-1, serum calcium, and 25(OH)D, which are summarized in Table 2.

A previous meta-analysis [19] by Dou et al. aimed to determine the effect of *n*-3 PUFA supplementation on bone mass in subjects over the age of 50. However, the study did not include several high-quality articles [29,30,57], including the VITAL trial [29], which had a large sample size of 771 subjects over 50 years old in America, investigating the effects of vitamin D3 and *n*-3 fatty acid supplements on BMD. In contrast, this meta-analysis and systematic review included trials that assessed BMD or bone markers as their primary goal, allowing for a comprehensive evaluation of the effects of *n*-3 PUFA on various measures of bone mineral density, bone markers, and other indexes of bone status. Additionally, this study attempted to provide a plausible mechanism for how *n*-3 PUFA affects bone health.

BMD is a crucial indicator of bone strength and is closely associated with bone health [64]. Following *n*-3 PUFA supplementation, an increase in blood *n*-3 PUFA levels was observed (Figure 6C), indicating the successful addition of *n*-3 PUFA to the body. As *n*-3 PUFA are known for their anti-inflammatory effects, changes in inflammatory factors (IL-6, CRP) are expected. However, the present study did not find any significant changes in inflammatory factors (Figure 6A,B), suggesting that *n*-3 PUFA may not have a significant effect on other indicators. Moreover, in the overall analysis, no significant effect of *n*-3 PUFA supplementation was found on any part of the body’s BMD (Figure 2). To analyze the effect of *n*-3 PUFA on BMD at different sites, BMD was divided into lumbar spine, femoral neck, and total hip bone mineral density. Following *n*-3 PUFA supplementation, individuals from Eastern countries exhibited higher BMD at the lumbar spine and femoral neck than those from Western countries (Table 2). This result could be attributed to dietary differences, as the Western diet is high in polyunsaturated fatty acids, and the American diet contains 11 to 25 times more *n*-6 fatty acids than *n*-3 fatty acids [65]. The *n*-3 PUFA:*n*-6 PUFA ratio is a valuable indicator of the balance between these two types of fatty acids in the body. *N*-3 PUFA and *n*-6 PUFA compete for the same enzymes involved in their metabolism and exert opposing effects on inflammatory pathways [66]. A review also discovered that *n*-3 and *n*-6 PUFA have opposing effects on the human body [67], implying that *n*-3 PUFA and *n*-6 PUFA may have a competitive relationship. As all trials did not alter the subjects’ daily diet, it is likely that they not only consumed more *n*-3 PUFA but also consumed a significant amount of *n*-6 PUFA. Thus, the effect of *n*-3 PUFA supplementation on BMD at different parts of the body may vary due to differences in *n*-3 PUFA intake across regions. Some studies in our analysis used ALA as a source of *n*-3 PUFA, which is a precursor to more active forms like DHA [68]. The variations observed in our study could be attributed to differences in ALA metabolism. However, further research is needed to understand the specific impact of ALA metabolism on our findings.

Another potential reason for the lack of statistical significance could be the effect of *n*-6 PUFA mitigating the effect of *n*-3 PUFA, in line with previous human and animal studies [40,69]. As previously stated, the Western diet already contains adequate amounts of *n*-3 PUFA, allowing individuals to consume enough *n*-3 PUFA without additional supplementation. Jørgensen et al. [40] discovered positive correlations between marine *n*-3 PUFA levels and Norwegian baseline Z-scores. Therefore, we hypothesized that if the proportion of *n*-3 PUFA was already above a certain level, additional supplementation may not provide any further benefit. Additionally, after receiving *n*-3 PUFA supplementation, women showed a greater positive effect on BMD at the femoral neck (Table 2). This may be because all four articles [54,58,59,62] included in the subgroup analysis were about postmenopausal women. A previous study on sex differences in the response to *n*-3 PUFA supplementation in the elderly found that *n*-3 PUFA supplementation increases muscle function and quality in older women but not in older men [70]. Therefore, the current study’s findings may be explained by the fact that women respond more positively to *n*-3 PUFA supplementation than men. However, further investigation is warranted to understand the effects of *n*-3 PUFA supplementation on bone in individuals with different dietary patterns and genders. Hormone replacement therapy (HRT) could be a potential factor contributing to the observed differences between sexes. HRT, especially estrogen therapy, has a significant impact on bone metabolism and characteristics [71]. Therefore, it is important to consider the potential influence of HRT when interpreting our findings. Future research should specifically investigate the effects of hormone replacement therapy to better understand its relationship with the interventions studied.

The degree and process of bone mineralization are primary factors that influence bone mineral density (BMD) [72]. The primary storage form of vitamin D in the body is 25(OH)D, and the serum 25(OH)D level may serve as an indicator of the level of vitamin D in the body to some extent [73]. Clinical and experimental studies have demonstrated that *n*-3 PUFA supplementation can significantly increase serum vitamin D levels [74,75]. Our subgroup analysis revealed that when *n*-3 PUFA supplementation was administered for less than six months, serum 25(OH)D levels significantly increased, while serum calcium concentration significantly decreased after six months (Table 2). These findings suggest that *n*-3 PUFA supplementation can enhance vitamin D absorption and utilization in the short term, but this effect diminishes over time. One possible explanation is that two [55,59] of the randomized controlled trials (RCTs) included in this study were conducted on postmenopausal healthy women. Previous research that compared the effects of cholecalciferol on 1-year changes in calcium absorption and BMD in postmenopausal women discovered that serum vitamin D levels were highest at 30 days and decreased at 60, 120, 240, and 365 days [76]. Therefore, we speculate that when changes in body hormone levels are relatively small, a fixed amount of *n*-3 PUFA supplementation can compensate for the decrease in vitamin D absorption rate caused by hormonal changes. However, over time, as hormone levels continue to change, the previous amount may become insufficient to compensate for the low vitamin D absorption rate over a long period of time. The current study found that long-term, *n*-3 PUFA supplementation reduces vitamin D absorption and utilization but has no effect on the dynamic balance of blood calcium and bone calcium. Interestingly, supplementing with *n*-3 PUFA improved serum 25(OH)D levels more in individuals over 60 years of age than in those under 60, indicating that age plays a significant role in the effectiveness of *n*-3 PUFA in regulating bone mineralization (Table 2). One possible explanation is that elderly individuals over 60 years of age exhibit increased bone resorption activity and weak bone formation, which inhibits the degree of bone mineralization. To improve bone mineralization in the elderly population, a significant amount of inorganic salt absorption, such as calcium, is required. Based on the above findings, we speculate that supplementing with *n*-3 PUFA may be more effective in elderly residents over the age of 60 in the short term, and further research is needed. The co-supplementation of *n*-3 PUFA and vitamin D in some studies may have contributed to a greater improvement in 25(OH)D levels, especially in individuals aged 60 and above [77,78]. However, further research is needed to determine the optimal dosage and combination of *n*-3 PUFA and vitamin D for promoting adequate 25(OH)D levels, particularly in the elderly population.

We conducted a comprehensive analysis of various biomarkers to investigate the potential metabolic pathways through which *n*-3 PUFA may impact bone health, but our results did not show any significant effects, which is consistent with the findings on BMD. Specifically, we examined biomarkers such as BAP and OC to assess whether *n*-3 PUFA have any impact on bone formation, as well as CTx-1 and NTx-1 to evaluate whether *n*-3 PUFA affect bone resorption in old or necrotic bone. Additionally, we investigated PTH, serum calcium, and 25(OH)D levels to determine if there were any changes following *n*-3 PUFA supplementation. However, none of these biomarkers showed any significant differences.

Our overall analyses were not sufficiently robust to draw a firm conclusion that *n*-3 PUFA have no effect on femoral neck BMD, serum calcium, or 25(OH)D, as the sensitivity analysis of these parameters showed some variations. Therefore, we must interpret these findings with caution. Consequently, we can only speculate that *n*-3 PUFA may have an indirect effect on maintaining the balance of bone metabolism.

Despite the promising results of this study, there are still several limitations that need to be addressed. First, moderate to high heterogeneity in trial results was found in several meta-analyses due to a few small-to-moderate-sized studies reporting positive results that were not seen in larger trials. Therefore, caution is advised when interpreting results from smaller studies, as they may not be replicated in larger or longer studies. Second, more large-scale randomized controlled trials are needed to provide more conclusive evidence on the effects of *n*-3 PUFA on bone health. Additionally, further research is necessary to determine the efficacy of combining *n*-3 PUFA supplements with other bone-beneficial substances. Rigorous studies with larger sample sizes and controlled designs should investigate specific combinations, dosages, and treatment durations to guide clinical recommendations for promoting bone health. Third, the findings of this study may have been influenced by the different methods used to collect data for OC, BAP, serum calcium, 25(OH)D, and PTH in different trials. Finally, the current study did not investigate the effects of *n*-3 PUFA on other bone-related outcomes, such as fracture risk or bone turnover markers, which may be important to consider in future research. Despite these limitations, this study provides valuable insights into the potential benefits of *n*-3 PUFA supplementation on bone health.

## 5. Conclusions

Based on our current analyses, there is no definitive evidence to support the significant improvement of bone health in adults through *n*-3 PUFA supplementation or to elucidate how *n*-3 PUFA affects bone metabolism. While our data suggest that *n*-3 PUFA supplementation may have a positive impact on BMD, 25(OH)D, serum calcium, and NTx-1 in elderly women from Eastern countries, these findings are tentative and require further investigation. Further high-quality studies with larger and diverse populations are needed to fully understand the impact of *n*-3 PUFA supplementation and combined supplementation of *n*-3 PUFA on bone health.

## Figures and Tables

**Figure 1 nutrients-15-02806-f001:**
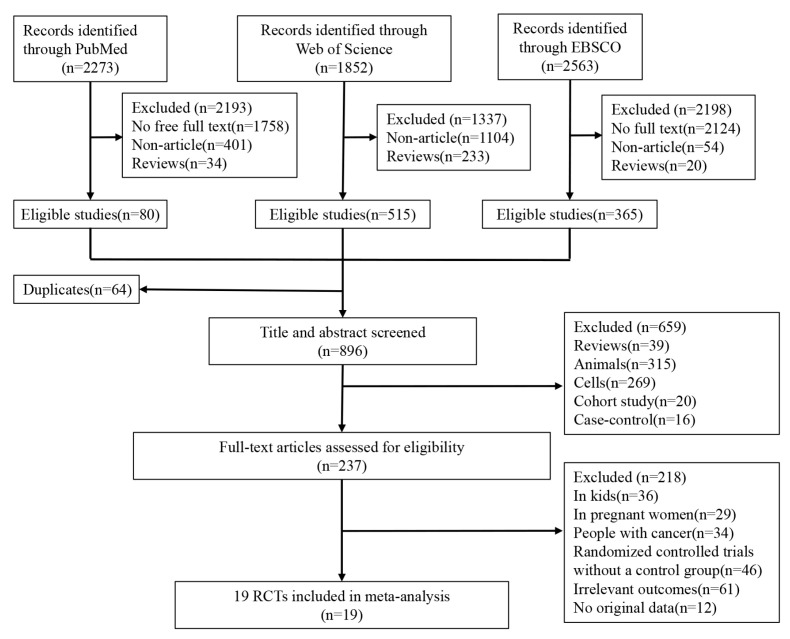
Study selection.

**Figure 2 nutrients-15-02806-f002:**
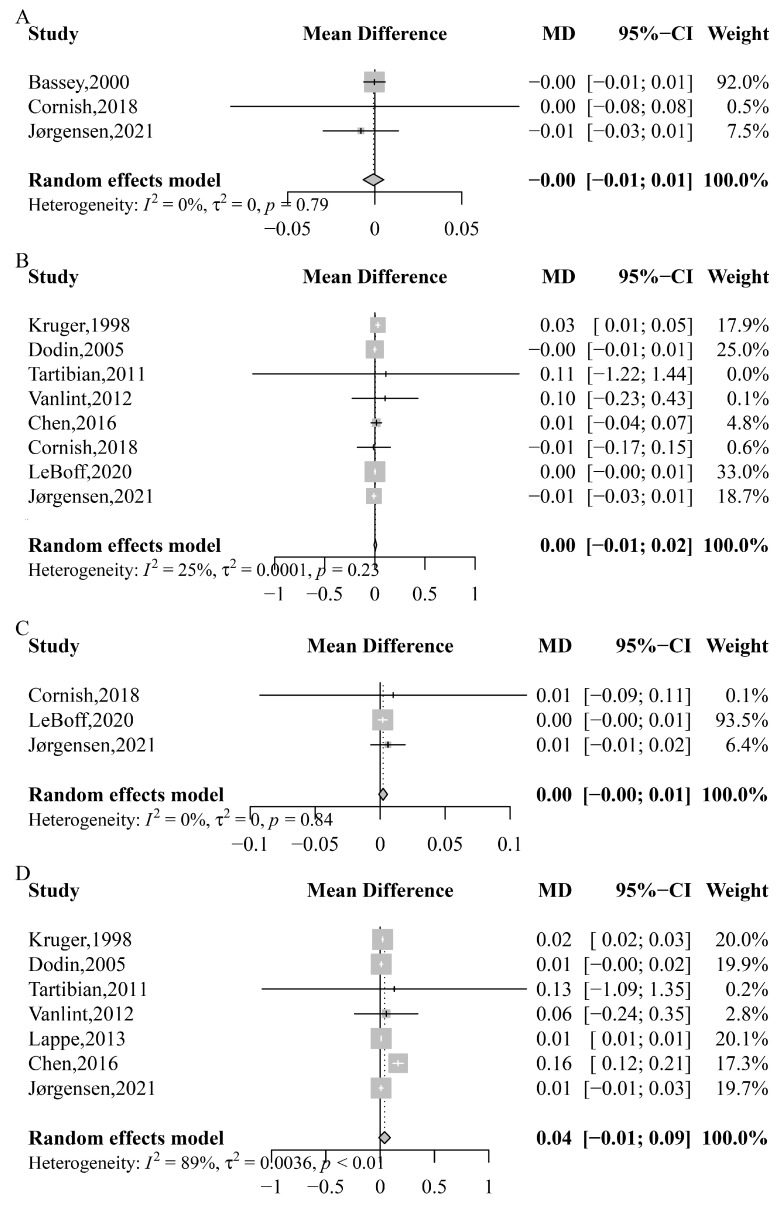
Meta-analysis of the effects of *n*-3 PUFA on different parts of the body bone mineral density. (**A**) Total body mineral density, Bassey, 2000 [51], Cornish, 2018 [39], Jørgensen, 2021 [40]; (**B**) lumbar spine bone mineral density, Kruger, 1998 [58], Dodin, 2005 [54], Tartibian, 2011 [62], Vanlint, 2012 [63], Chen, 2016 [52], Cornish, 2018 [39], LeBoff, 2020 [29], Jørgensen, 2021 [40]; (**C**) total hip bone mineral density, Cornish, 2018 [39], LeBoff, 2020 [29], Jørgensen, 2021 [40]; (**D**) femoral neck bone mineral density, Kruger, 1998 [58], Dodin, 2005 [54], Tartibian, 2011 [62], Vanlint, 2012 [63], Chen, 2016 [52], Jørgensen, 2021 [40].

**Figure 3 nutrients-15-02806-f003:**
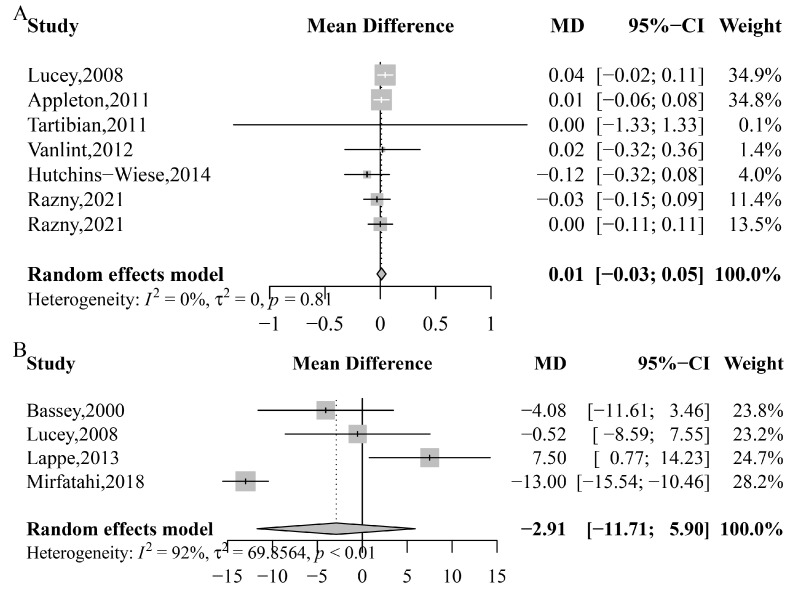
Meta-analysis of the effects of *n*-3 PUFA on bone absorption markers. (**A**) *C*-terminal telopeptide of type I collagen (CTx-1), Lucey, 2008 [60], Appleton, 2011 [50], Tartibian, 2011 [62], Vanlint, 2012 [63], Hutchins-Wiese, 2014 [56], Razny, 2021 [41], Razny, 2021 [41]; (**B**) *N*-telopeptides of type I collagen (NTx-1), Bassey, 2000 [51], Lucey, 2008 [60], Lappe, 2013 [59], Mirfatahi, 2018 [30].

**Figure 4 nutrients-15-02806-f004:**
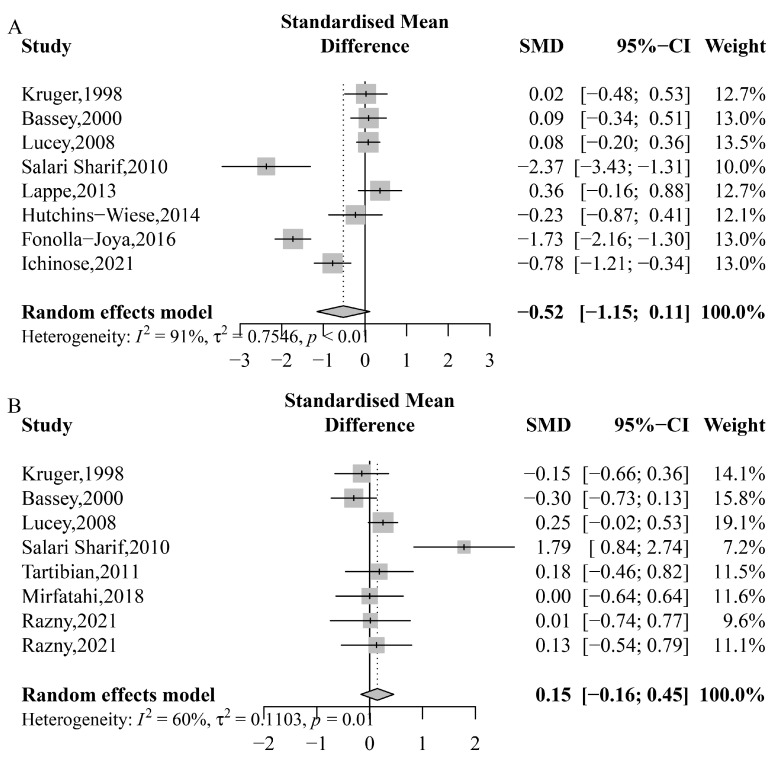
Meta-analysis of the effects of *n*-3 PUFA on bone formation biomarkers. (**A**) Bone-specific alkaline phosphatase (BAP), Kruger, 1998 [58], Bassey, 2000 [51], Lucey, 2008 [60], Salari Sharif, 2010 [61], Lappe, 2013 [59], Hutchins-Wiese, 2014 [56], Fonolla-Joya, 2016 [55], Ichinose, 2021 [57]; (**B**) osteocalcin (OC), Kruger, 1998 [58], Bassey, 2000 [51], Lucey, 2008 [60], Salari Sharif, 2010 [61], Tartibian, 2011 [62], Mirfatahi, 2018 [30], Razny, 2021 [41], Razny, 2021 [41].

**Figure 5 nutrients-15-02806-f005:**
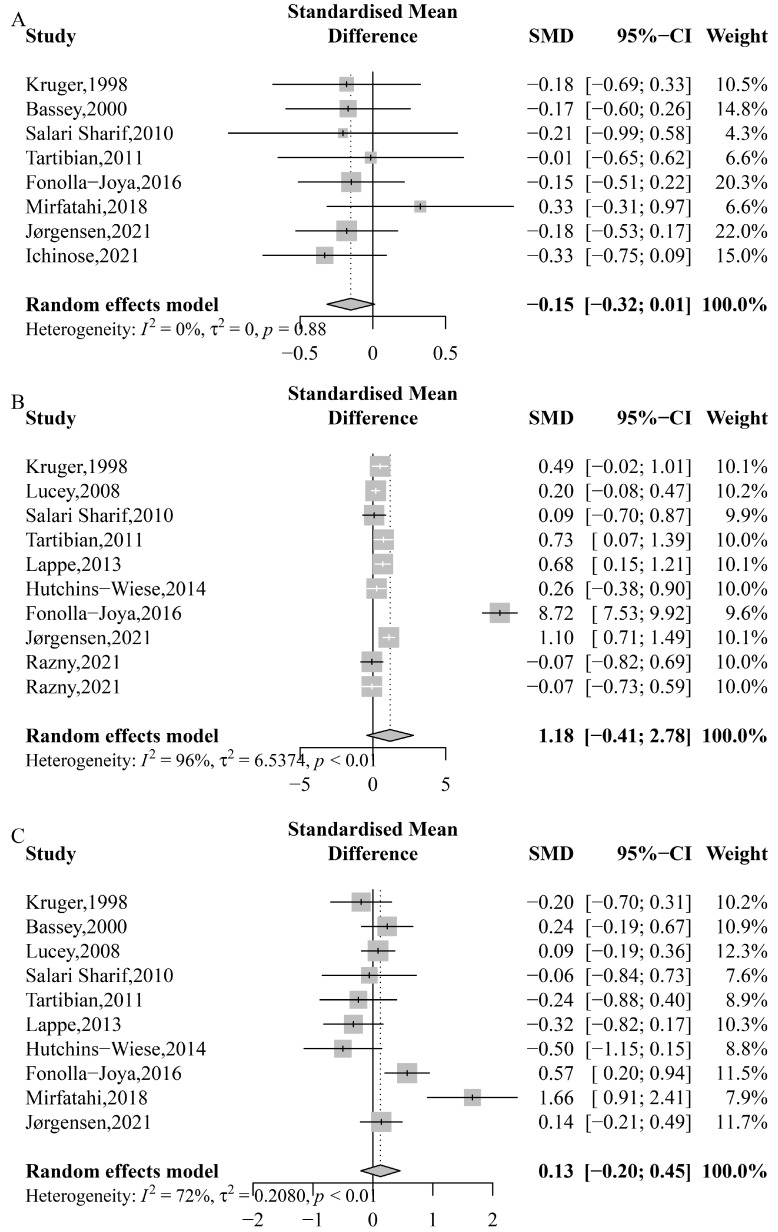
Meta-analysis of the effects of *n*-3 PUFA on serum calcium, 25(OH)D, and PTH. (**A**) Serum calcium, Kruger, 1998 [58], Bassey, 2000 [51], Salari Sharif, 2010 [61], Tartibian, 2011 [62], Fonolla-Joya, 2016 [55], Mirfatahi, 2018 [30], Jørgensen, 2021 [40], Ichinose, 2021 [57]; (**B**) 25(OH)D, Kruger, 1998 [58], Lucey, 2008 [60], Salari Sharif, 2010 [61], Tartibian, 2011 [62], Lappe, 2013 [59], Hutchins-Wiese, 2014 [56], Fonolla-Joya, 2016 [55], Jørgensen, 2021 [40], Razny, 2021 [41], Razny, 2021 [41]; (**C**) parathyroid hormone (PTH), Kruger, 1998 [58], Bassey, 2000 [51], Lucey, 2008 [60], Salari Sharif, 2010 [61], Tartibian, 2011 [62], Lappe, 2013 [59], Hutchins-Wiese, 2014 [56], Fonolla-Joya, 2016 [55], Mirfatahi, 2018 [30], Jørgensen, 2021 [40].

**Figure 6 nutrients-15-02806-f006:**
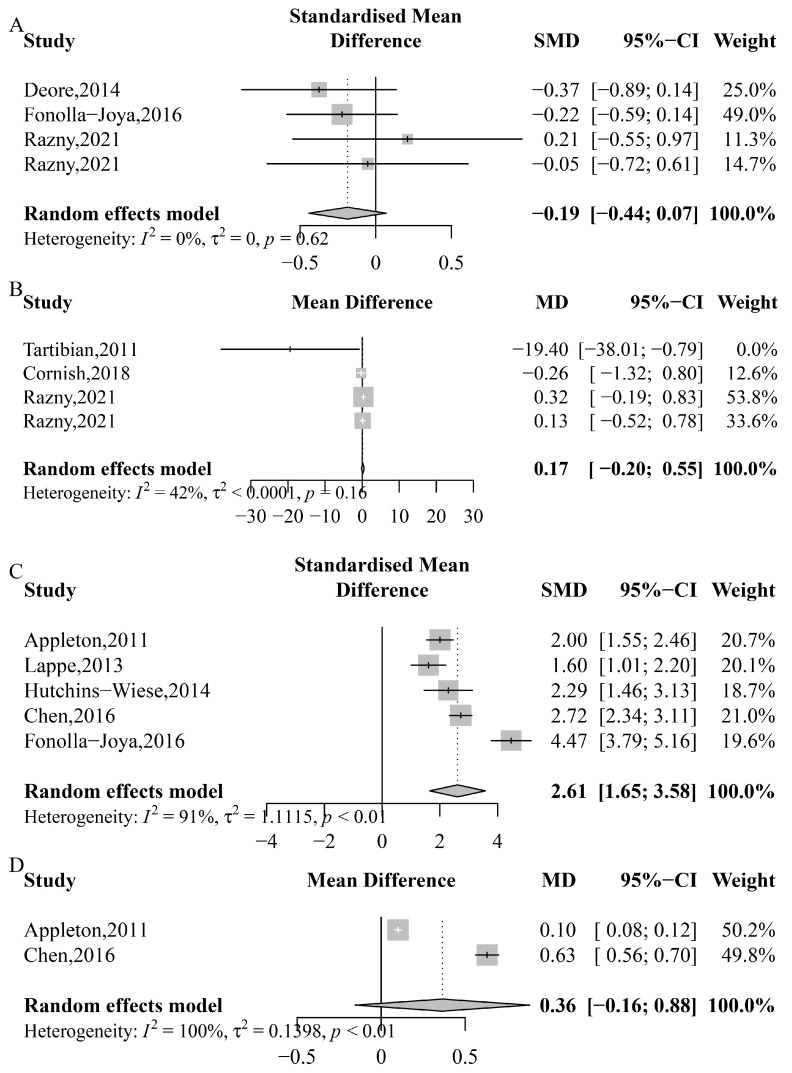
Meta-analysis of the effects of *n*-3 PUFA on CRP, IL-6, blood *n*-3 PUFA, and blood *n*-3 PUFA:*n*-6 PUFA ratio. (**A**) *C*-reactive protein (CRP), Deore, 2014 [53], Fonolla-Joya, 2016 [55], Razny, 2021 [41], Razny, 2021 [41]; (**B**) interleukin-6 (IL-6), Tartibian, 2011 [62], Cornish, 2018 [39], Razny, 2021 [41], Razny, 2021 [41]; (**C**) blood *n*-3 PUFA, Appleton, 2011 [50], Lappe, 2013 [59], Hutchins-Wiese, 2014 [56], Chen, 2016 [52], Fonolla-Joya, 2016 [55]; (**D**) blood *n*-3 PUFA:*n*-6 PUFA ratio, Appleton, 2011 [50], Chen, 2016 [52].

**Table 1 nutrients-15-02806-t001:** Characteristics of included studies.

First Author (Year)	Nation	G	Age (Years)	Population	N (I/C)	I/C	D	Outcomes	Modified Jadad Scale Score
Kruger (1998) [58]	South Africa	F	79.5 ± 5.56	osteopenic women	60 (29/31)	0.42 g *n*-3/placebo (SFA)	18 mo	sCa; 25(OH)D; BAP; OC; PTH; lumbar spine BMD; femoral neck BMD	3
Bassey (2000) [51]	UK	F	45.7 ± 11.1	pre- and post-menopausal women	88 (40/48)	Efacal/Ca	12 mo	sCa; NTx-1; BAP; OC; PTH; TBMD	5
Dodin (2005) [54]	Canada	F	54.7 ± 4.3	women aged 45–65	179 (85/94)	flaxseed oil (9.12 g *n*-3)/placebo (wheat germ)	12 mo	lumbar spine BMD; femoral neck BMD	7
Lucey (2008) [60]	US	M&F	32.5 ± 5.3	young overweight adults	276 (210/66)	lean fish (0.3 g *n*-3), salmon (3.0 g *n*-3), fish-oil (1.5 g *n*-3)/control (no seafood)	2 mo	25(OH)D; NTx-1; CTx-1; BAP; OC; PTH	4
Salari Sharif (2010) [61]	Iran	F	61.4 ± 7.4	postmenopausal osteoporotic women	25 (13/12)	*n*-3 capsules (0.9 g *n*-3)/placebo	6 mo	sCa; 25(OH)D; BAP; OC; PTH	4
Appleton (2011) [50]	UK	M&F	NA	mild–moderately depressed individuals	113 (53/60)	1.48 g *n*-3 (0.63 g EPA + 0.85 g DHA)/placebo (olive oil)	3 mo	CTx-1; *n*-3 PUFA: *n*-6 PUFA; serum *n*-3 PUFA	5
Tartibian (2011) [62]	Iran	F	61.1 ± 8.0	healthy postmenopausal women aged 58–78	38 (20/18)	*n*-3 capsules (0.6 g EPA + 0.4 g DHA)/ control	6 mo	sCa; 25(OH)D; CTx-1; OC; PTH; lumbar spine BMD; femoral neck BMD; IL-6	3
Vanlint (2012) [63]	Australia	M&F	59.2	individuals with osteopenia	40 (20/20)	0.4 g DHA + Ca/placebo (Ca)	12 mo	CTx-1; lumbar spine BMD; femoral neck BMD	6
Lappe (2013) [59]	US	F	54.7 ± 2.4	postmenopausal women	70 (35/35)	GBB/control (Ca)	6 mo	25(OH)D; NTx-1; BAP; PTH; femoral neck BMD; serum *n*-3 PUFA	6
Hutchins (2014) [56]	US	F	61.8 ± 9.5	postmenopausal women	38 (20/18)	4.0 g n3 (EPA + DHA)/placebo (sunflower oil)	3 mo	25(OH)D; CTx-1; BAP; PTH; serum *n*-3 PUFA	6
Deore (2014) [53]	India	NA	44.9 ± 28.9	patients with chronic periodontitis aged 30–60	60 (30/30)	0.18 g EPA + 0.12 g DHA/placebo (liquid paraffin)	3 mo	CRP	6
Chen (2016) [52]	Australia	M&F	61.0 ± 10.0	aged over 40 years with clinical knee OA	202 (101/101)	high dose (4.5 g EPA + 4.5 g DHA); low dose (0.45 g EPA + 0.45 g DHA)	24 mo	lumbar spine BMD; femoral neck BMD; *n*-3 PUFA: *n*-6 PUFA; serum *n*-3 PUFA	7
Fonolla-Joya (2016) [55]	Spain	F	59.7 ± 5.8	postmenopausal women	117 (63/54)	0.2 g n3 (EPA + DHA) + vitamins/control(vitamins)	12 mo	sCa; 25(OH)D; BAP; PTH; serum *n*-3 PUFA; CRP	5
Mirfatahi (2018) [30]	Iran	M&F	63.5 ± 5.7	stable hemodialysis patients	38 (19/19)	flaxseed oil (3.45 g ALA)/control (triglycerides oil)	2 mo	sCa; NTx-1; OC; PTH	7
Cornish (2018) [39]	Canada	M	71.1 ± 5.5	men ≥65	24 (12/12)	3.0 g n3 (EPA + DHA)/control (omega 3-6-9)	3 mo	TBMD; lumbar spine BMD; total hip BMD; IL-6	4
LeBoff (2020) [29]	US	M&F	63.8 ± 6.1	men ≥50, women ≥55	771 (388/383)	VtD + 1.0 g n3/placebo	24 mo	lumbar spine BMD; total hip BMD	4
Jørgensen (2021) [40]	Norway	M&F	53 ± 14	recipients with stable kidney function	132 (66/66)	2.6 g n3 (0.46 g EPA + 0.38 g DHA)/placebo (olive oil)	11 mo	sCa; 25(OH)D; PTH; TBMD; lumbar spine BMD; total hip BMD; femoral neck BMD	5
Razny (2021) [41]	Poland	M&F	41.0 ± 9.9	subjects aged 25–65 with overweight or abdominal obesity	189 (95/94)	IS + 1.8 g n3; CR + 1.8 g n3/IS + placebo; CR + placebo	3 mo	25(OH)D; CTx-1; OC; CRP; IL-6	6
Ichinose (2021) [57]	Japan	M&F	69.1 ± 5.1	healthy elderly Japanese	87 (46/41)	acid-enriched milk (0.434 g n3)/placebo milk	12 mo	sCa; BAP	6

Abbreviations: UK, United Kingdom; US, United States; F, female; M, male; NA, not available; G, gender; N, number of total sample size; I, intervention; C, control; D, Duration; SFA, saturated fatty acids; n3, *n*-3 PUFA; Efacal, Ca 1.0 g/d + evening primrose oil 4.0 g/d+ marine fish oil 0.44 g/d; GBB, geniVida bone blend (genistein 30 mg + VtD_3_ 800 IU + 1 g n3 PUFA (EPA:DHA ≈ 2:1)); ALA, α-linolenic acid; IS, isocaloric diet; CR, low-calorie diet; VtD, vitamin D; sCa, serum calcium; BAP, bone-specific alkaline phosphatase; OC, osteocalcin; PTH, parathyroid hormone; BMD, bone mineral density; NTx-1, *N*-telopeptides of type I collagen; TBMD, total body mineral density; CTx-1, *C*-terminal telopeptide of type I collagen; PUFA, polyunsaturated fatty acids; IL-6, interleukin-6; CRP, *C*-reactive protein.

**Table 2 nutrients-15-02806-t002:** Subgroup analyses with significant differences.

Subgroup Analysis	Lumbar Spine BMD	Femoral Neck BMD	NTx-1	Serum Calcium	25(OH)D	Blood *n*-3 PUFA
Country						
Eastern country	0.03 [0.01, 0.05]	0.02 [0.02, 0.03]				
Western country	0.00 [−0.00, 0.01]	0.05 [−0.02, 0.11]				
Gender						
male		0.01 [0.01, 0.02]	1.10 [−5.73, 7.94]			2.79 [1.08, 4.49]
female and male		0.08 [−0.04, 0.20]	−9.66 [−15.60, −3.71]			2.38 [1.67,3.08]
Average age class						
≥60		0.09 [−0.04, 0.23]			0.43 [0.11, 0.74]	2.65 [2.30, 3.00]
<60		0.01 [0.01, 0.01]			1.52 [−0.79, 3.84]	3.03 [0.22, 5.84]
Control type						
placebo		0.02 [0.01, 0.03]				2.07 [1.67, 2.47]
other control		0.09 [−0.06, 0.23]				2.92 [1.31, 4.54]
Duration class						
>6 months		0.05 [−0.02, 0.12]		−0.19 [−0.37, −0.01]	3.41 [−1.75, 8,57]	3.57 [1.86, 5.29]
≤6 months		0.01 [0.01, 0.01]		0.07 [−0.33, 0.46]	0.29 [0.11, 0.47]	1.93 [1.59, 2.26]
Intervention type						
*n*-3 only					0.36 [0.06, 0.66]	2.36 [1.87, 2.85]
*n*-3 + others					3.28 [−2.02, 8.57]	3.03 [0.22, 5.84]
Dose class						
>1 g/d						2.36 [1.87, 2.85]
≤1 g/d						3.03 [0.22, 5.84]

Data expressed as pooled effect size [95% CI]. BMD, bone mineral density; NTx-1, *N*-telopeptides of type I collagen; PUFA, polyunsaturated fatty acids. Detailed forest plots of subgroup analyses were shown in Appendix A.

## Data Availability

The data presented in this study are available on reasonable request from the corresponding author.

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
