# Peer review of "The Effects of *n*-3 PUFA Supplementation on Bone Metabolism Markers and Body Bone Mineral Density in Adults: A Systematic Review and Meta-Analysis of RCTs"

_nutrients, 2023, doi:10.3390/nu15122806_

Round 1

Reviewer 1 Report

The meta-analysis conducted by Gao et al. shows that n-3 PUFA supplementation seems to have no significant effects on bone mineral density or bone metabolism. This study is well-designed, controlled, and written. There are some points that authors can consider adjusting their results: Obesity or the presence of diabetes can affect bone health, and it is related to lower levels of n-3 PUFA, then it would be desirable if authors may explore the effect of these variables. The possible confounding effect of therapy replacement hormone should be considered in women because the studies included pre- and post-menopausal women, which can explain the variation between the sexes. Also, a couple of studies included supplementation based on ALA sources rather than fish oil. Then, some differences due to the metabolism of ALA can be analyzed or discussed because ALA is a precursor of more active n-3 PUFA (DHA).

Reviewer 2 Report

The aim of this study was to investigate the effects of n-3 PUFA supplementation on markers of bone metabolism and bone mineral density through a meta-analysis of randomized controlled trials. The introduction lacks an explanation of markers for bone metabolism and mineral density.  In the Methods the authors wrote: "We analyzed the changes in bone mineral density (BMD) at different sites, bone markers, serum calcium, vitamin D, parathyroid hormone, n-3 PUFA, n-3 PUFA:n-6 PUFA ratio, and inflammatory markers", please explain why these biomarkers were used (lines 112-114). How BMD was measured? I also miss the reasoning how to link n-3 - inflammatory biomarkers - bone metabolism. 

Methodologically, the manuscript is generally very correct, except for not narrowing down the selection criteria. In my opinion, due to the high dependence on age, genotypes, habitual diet, use of non specific biomarkers, the criteria for selecting publications are too broad and need to be narrowed down.

Also, it can be assumed that n-3 foods and supplements usually contain vitamin D as well, and perhaps the n-3 relationship does not apply.

The studies included in the analysis (Table 1) do not concern only n-3 supplementation, so either the title, aim and description need to be changed or the analysis of the results
